# Depleting Trim28 in adult mice is well tolerated and reduces levels of α-synuclein and tau

Maxime WC Rousseaux[1,2†], Jean-Pierre Revelli[1,2], Gabriel E Vázquez-Velez[2,3,4], Ji-Yoen Kim[1,2], Evelyn Craigen[1,2], Kristyn Gonzales[1,2], Jaclyn Beckinghausen[2,5], Huda Y Zoghbi[1,2,3,5,6]*

[1]Department of Molecular and Human Genetics, Baylor College of Medicine, Houston, United States; [2]Jan and Dan Duncan Neurological Research Institute at Texas Children's Hospital, Houston, United States; [3]Program in Developmental Biology, Baylor College of Medicine, Houston, United States; [4]Medical Scientist Training Program, Baylor College of Medicine, Houston, United States; [5]Department of Neuroscience, Baylor College of Medicine, Houston, United States; [6]Howard Hughes Medical Institute, Baylor College of Medicine, Houston, United States

*For correspondence:
hzoghbi@bcm.edu

Present address: †Department of Cellular and Molecular Medicine, University of Ottawa Brain and Mind Research Institute, University of Ottawa, Ontario, Canada

**Abstract** Alzheimer's and Parkinson's disease are late onset neurodegenerative diseases that will require therapy over decades to mitigate the effects of disease-driving proteins such tau and α-synuclein (α-Syn). Previously we found that TRIM28 regulates the levels and toxicity of α-Syn and tau (*Rousseaux et al., 2016*). However, it was not clear how TRIM28 regulates α-Syn and it was not known if its chronic inhibition later in life was safe. Here, we show that TRIM28 may regulate α-Syn and tau levels via SUMOylation, and that genetic suppression of Trim28 in adult mice is compatible with life. We were surprised to see that mice lacking Trim28 in adulthood do not exhibit behavioral or pathological phenotypes, and importantly, adult reduction of TRIM28 results in a decrease of α-Syn and tau levels. These results suggest that deleterious effects from TRIM28 depletion are limited to development and that its inhibition adulthood provides a potential path for modulating α-Syn and tau levels.

DOI: https://doi.org/10.7554/eLife.36768.001

## Introduction

Neurodegenerative disorders such as Alzheimer's disease (AD) and Parkinson's disease (PD) occur in the later decades of life and have no curative therapy. Therefore, future treatments for these disorders must be administered over decades, which means that safety profiles of therapeutic targets are of utmost importance. The advent of alternative therapies such as antisense oligonucleotides, gene therapy and immunotherapy, together with traditional pharmacology have made it such that almost any molecule can be targeted. More and more, the extent to which a target is druggable hinges on the safety and specificity of its targeting over time.

We recently demonstrated that TRIM28 regulates the steady state levels of the neurodegeneration-driving proteins α-Synuclein (α-Syn) and tau (*Rousseaux et al., 2016*). However, given the critical roles of TRIM28 in mammalian development (*Cammas et al., 2000*), its tractability as a therapeutic target remains questionable. For instance, complete loss of *Trim28* in mice causes early embryonic lethality due to pre-implantation defects (*Cammas et al., 2000*), and specific deletion of this gene in the developing tissues cause a host of defects (*Cheng et al., 2014*; *Fasching et al.,*

2015; *Trono, 2015*). Moreover, haploinsufficiency of *TRIM28* is expected to have deleterious outcomes in humans (pLI = 1.00, ExAC; [*Lek et al., 2016*]). This may be due in part due to the multiple functions of TRIM28 within the cell including the repression of endogenous retroviral elements, maintenance of pluripotency, epigenetics and mitophagy (*Barde et al., 2013*; *Czerwińska et al., 2017*; *Oleksiewicz et al., 2017*; *Singh et al., 2015*; *Wolf and Goff, 2009*). Given the importance of TRIM28 for development, it remains unclear whether TRIM28 is critical for adult brain function, and whether it may safely be targeted in adulthood. Specifically, two questions remain related to the targeting of TRIM28 pharmacologically: (1) Is there a pharmacologically tractable domain in TRIM28 that could be targeted by a drug? (2) Is genetic suppression of TRIM28 in the brain and throughout the body tolerated in adulthood? To test this, we performed studies to pinpoint the mechanism by which TRIM28 regulates α-Syn and tau and generated two animal models to disrupt Trim28 in vivo, thus establishing its druggability in adulthood.

## Results and discussion

We previously found that TRIM28 regulates the post-translational stability of α-Syn and tau and that this effect is mediated by two critical cysteines in its RING domain (C65 and C68; [*Rousseaux et al., 2016*]). We hypothesized that TRIM28 may act as an E3 SUMO ligase (*Liang et al., 2011*; *Neo et al., 2015*; *Yang et al., 2013*) toward α-Syn and tau via this domain for three reasons: (1) TRIM28 interacts only weakly with α-Syn and tau (*Rousseaux et al., 2016*) and is therefore unlikely to act solely as a stabilizing factor via these residues; (2) TRIM28 mediates the nuclear localization of α-Syn and tau and SUMOylation is thought to play a critical role in influencing subcellular localization (*Hay, 2005*); and (3) Given the post-translational stabilization effect of TRIM28 on α-Syn and tau (*Rousseaux et al., 2016*), we surmised that SUMOylation may help prevent polyubiquitination, thus increasing their overall bioavailability. To test whether SUMOylation itself regulates the levels of α-Syn and tau, we inhibited the sole E2 SUMO ligase, UBC9, via RNAi and pharmacological inhibition (using Viomellein [*Hirohama et al., 2013*]). We found that both approaches were sufficient to decrease α-Syn and tau, suggesting that SUMOylation indeed regulates their steady state levels (*Figure 1A*). We next asked whether TRIM28 mediates the SUMOylation of α-Syn and tau. We first tested this in cells and found that knockdown of endogenous TRIM28 decreased native α-Syn and tau SUMOylation whereas ectopic overexpression of TRIM28 increased their SUMOylation (*Figure 1B*). Interestingly, when we mutated a catalytic RING domain of TRIM28 (C65A/C68A), we could inhibit α-Syn and tau SUMOylation (*Figure 1B*). This was consistent with our previous findings that mutating this residue impeded α-Syn and tau stabilization and nuclear localization (*Rousseaux et al., 2016*). However, mutant TRIM28 (C65A/C68A) was less stable than its wildtype form in all assays. Thus, whether these findings are due to decreased SUMOylation or to a change in TRIM28 protein stability is difficult to discern. To further test whether Trim28 regulates α-Syn and tau SUMOylation, we performed SUMOylation assays on endogenous α-Syn and tau from brain lysates (under denaturing conditions) from wild-type and $Trim28^{+/-}$ mice. We found that α-Syn and tau SUMOylation were significantly reduced in *Trim28* haploinsufficient mice (*Figure 1C*).

TRIM28 has several important functions throughout the cell (*Cheng et al., 2014*; *Czerwińska et al., 2017*; *Dalgaard et al., 2016*; *Fasching et al., 2015*; *Liang et al., 2011*; *Neo et al., 2015*; *Singh et al., 2015*), and its loss of function in mice is embryonic lethal (*Cammas et al., 2000*). We asked whether one of its domains can be specifically targeted for future therapeutic use without disrupting the others. Given that two conserved critical cysteine residues in its RING domain (*Figure 1—figure supplement 1A*) regulate TRIM28 function toward α-Syn and tau, we hypothesized that mutating residues critical for its endogenous catalytic activity would be the most promising approach. We therefore generated a knockin mouse carrying mutations in its RING domain (*Figure 1—figure supplement 1B*). We found that mutating these residues, despite decreasing α-Syn and tau levels significantly, caused a dramatic destabilization of TRIM28 protein (*Figure 1—figure supplement 1C–D*). Moreover, homozygosity for the these E3 mutant allele caused embryonic lethality, a feature consistent with the effects of a null allele. Thus, mutating the RING domain of TRIM28 decreases α-Syn and tau levels, but does so by disrupting its structure and stability (*Figure 1—figure supplement 1D*).

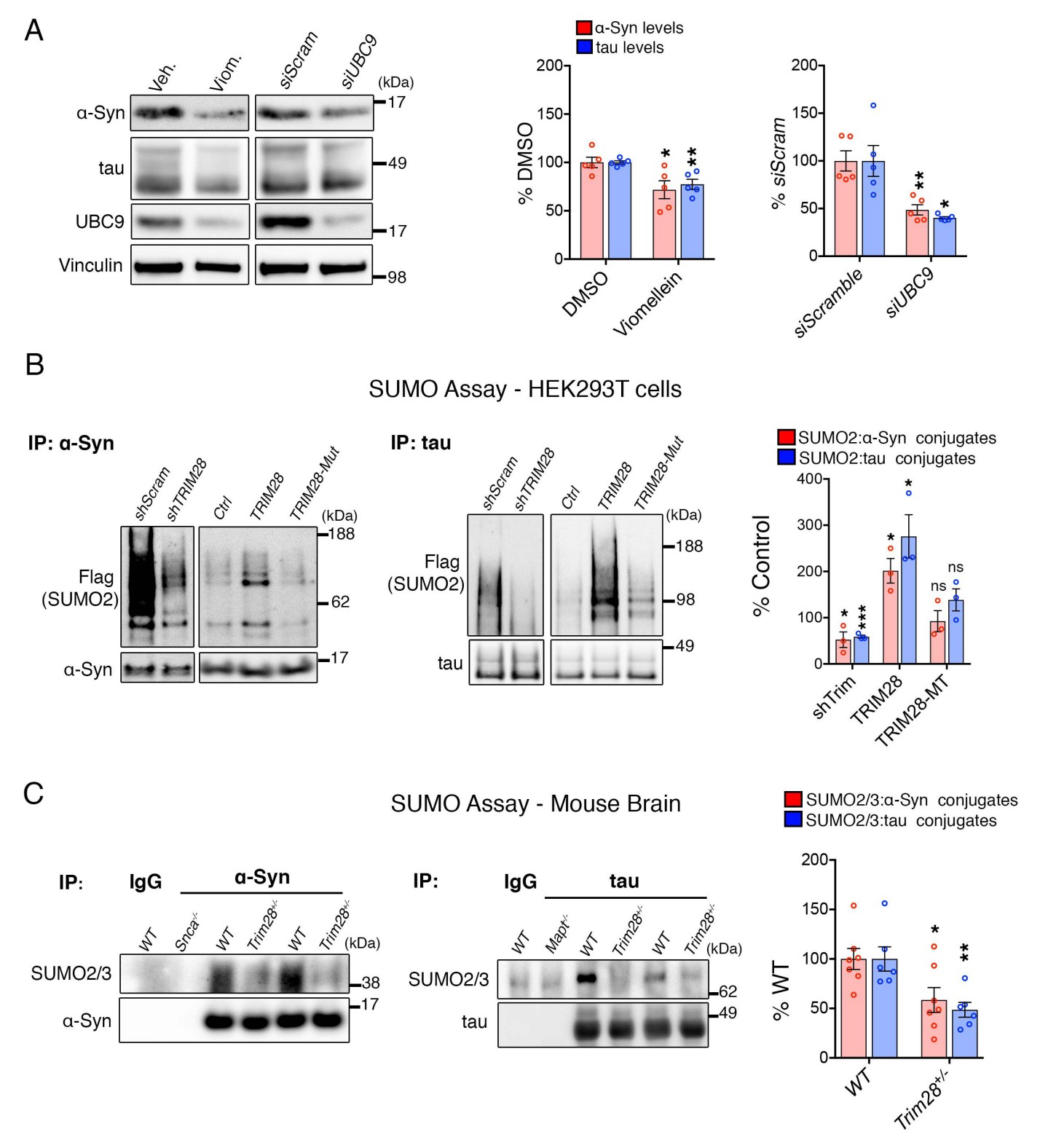

**Figure 1.** Trim28 mediates the SUMOylation of α-Syn and tau. (**A**) Blocking SUMOylation – by either pharmacological inhibition using viomellein or siRNA-mediated suppression of the sole SUMO E2 ligase, UBC9 – decreases α-Syn and tau levels by western blot. (**B**) SUMO assay in human cells reveals that TRIM28 mediates the formation of SUMO2 adducts on α-Syn and tau. This effect is lost upon mutation of the RING domain of TRIM28

*Figure 1 continued on next page*

*Figure 1 continued*

(TRIM28-Mut). (C) In vivo SUMO assay from denatured mouse brain lysates of WT and *Trim28*$^{+/-}$ mice. *Snca*$^{-/-}$ and *Mapt*$^{-/-}$ mice and IP: IgG serve as negative controls. \*, \*\*, \*\*\* and ns denote $p < 0.05$, $p < 0.01$, $p < 0.001$ and $p > 0.05$, respectively.

DOI: https://doi.org/10.7554/eLife.36768.002

The following figure supplement is available for figure 1:

**Figure supplement 1.** Ablating endogenous Trim28 catalytic activity dramatically reduces its stability, concomitantly decreasing α-Syn and tau levels.

DOI: https://doi.org/10.7554/eLife.36768.003

Since TRIM28 has critical roles in development, we next asked whether we could bypass these defects by knocking down Trim28 in the postnatal mouse brain (*Figure 2—figure supplement 1A*). We used an AAV carrying both an shRNA targeting *Trim28* and a YFP reporter. We found that the virus was widely expressed throughout the brain (*Kim et al., 2013*) and that mice receiving an shRNA against *Trim28* had a 75% depletion of *Trim28* in their brain (*Figure 2—figure supplement 1B*). Importantly, these mice developed normally until at least 10 weeks of age. We evaluated cortical and hippocampal thickness and astrocytosis in these mice and did not note any significant defects (*Figure 2—figure supplement 1C*).

Given that synucleinopathies and tauopathies most often occur in the later decades of life, therapeutics should therefore accurately mimic this late-stage disruption. To test whether late stage inhibition of Trim28 is therapeutically tractable, we generated Trim28 adult knockout mice. This was done by crossing a whole body, tamoxifen-inducible Cre (*UBC-CreER*$^{T2}$, [*Ruzankina et al., 2007*]) with mice carrying a floxed *Trim28* allele (*Cammas et al., 2000*). We waited until the animals were 8–12 weeks old before starting a 4 week tamoxifen regimen to ablate *Trim28* (*Figure 2A*). To our surprise, we found that adult depletion did not result in early lethality nor overt phenotypes. Instead, adult knockout mice lived for the duration of the study (over 40 weeks post-tamoxifen injection, *Figure 2B*). We tested whether Trim28 is effectively ablated in these mice and found that Trim28 levels were reduced by over 75% in each tissue tested (both at the RNA and protein level; *Figure 2C,D* and *Figure 2—figure supplement 2A–C*). Importantly, α-Syn and tau levels were also decreased in multiple brain regions, corroborating our previous findings using germline haploinsufficient mice (*Rousseaux et al., 2016*).

An important aspect of measurable safety margins in the depletion of a gene is its impact on neuronal function. To assess whether loss of Trim28 in adult mice impacts brain structure and function, we performed a battery of behavioral and histological tests. We found that Trim28 adult knockout mice behaved similarly to their control littermate counterparts in every test assayed. Specifically, no defects were observed in motor behavior, anxiety, perseverative movements and memory (*Figure 3A–H*). Consistent with this, we could not discern any gross histological defects nor signs of inflammation (as measured by GFAP immunoreactivity) in the brain (*Figure 4A–C*). We further tested Trim28 levels via immunostaining and found that, while Trim28 was highly expressed in the brain (confirming our western and qPCR results), it was depleted in the adult knockout (*Figure 4—figure supplement 1A*). A previous study highlighted several gene expression changes in mice lacking Trim28 in forebrain excitatory neurons starting from postnatal day 14 (*Jakobsson et al., 2008*). We tested the expression of these genes in the hippocampus using qPCR and found that, while the directionality of changes was consistent with the previous study, there was a broad dampening of this effect in the adult knockout mice (*Figure 4—figure supplement 1B*). This may be due to the later stage depletion of Trim28 or the incomplete deletion of Trim28 (there is 15–20% remaining in most adult knockouts) and may account for the slight behavioral abnormalities observed in the reported juvenile forebrain-specific Trim28 knockouts (*Jakobsson et al., 2008*) versus the whole-body adult Trim28 knockouts.

Given that the adult knockout affects the whole body, we examined regions of the body that could be vulnerable to Trim28 loss-of-function-induced toxicity. We assessed general morphology of the heart, liver and spleen and found no discernable defects in the adult knockout mice compared to littermate controls (*Figure 4—figure supplement 2*). Moreover, blood chemistry in these mice appeared normal (*Figure 4—figure supplement 3*).

Taken together, our study suggests that adult depletion of more than 75% of total Trim28 from the mouse body does not result in overt neurobehavioral phenotypes nor does it cause gross

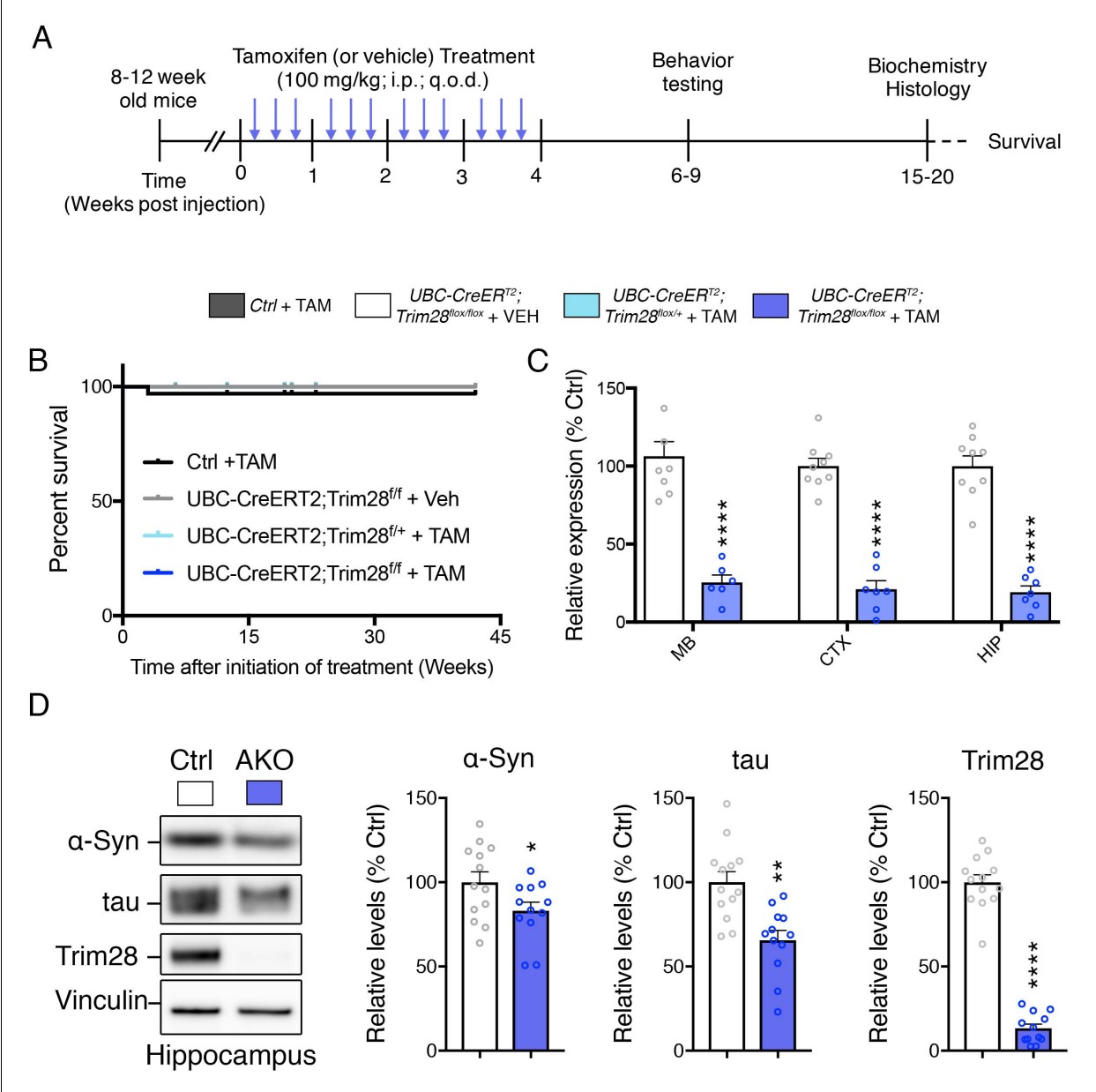

**Figure 2.** Trim28 adult knockout mice are viable and demonstrate reduced α-Syn and tau levels. (**A**) Experimental approach to delete Trim28 from the adult body. (**B**) Kaplan-Meier survival curve of Adult knockout mice (*UBC-CRE^ERT2; Trim28^flox/flox* + TAM vs littermate controls). No significant differences in survival are observed. (**C**) qPCR analysis for Trim28 expression in midbrain (MB), cortex (CTX) and hippocampus (HIP) of Trim28 adult knockout mice and control littermates. (**D**) Western blot analysis of α-Syn, tau and Trim28 levels in hippocampi from Trim28 adult knockout mice and control littermates. In (**B**), *n* = 14–33 per group. In (**C** and **D**), *n* = 12–13 per group.

DOI: https://doi.org/10.7554/eLife.36768.004

The following figure supplements are available for figure 2:

**Figure supplement 1.** Perinatal suppression of Trim28 in the brain is safe and decreases α-Syn and tau levels.

DOI: https://doi.org/10.7554/eLife.36768.005

**Figure supplement 2.** α-Syn and tau levels are reduced in multiple brain regions from Trim28 adult knockout mice.

DOI: https://doi.org/10.7554/eLife.36768.006

histological or biochemical defects. This is consistent with reports that deletion of TRIM28 in termi-nally differentiated muscle does not cause obesity (***Dalgaard et al., 2016***). These findings hold important implications for therapeutic targeting of Trim28 in diseases such as AD and PD where an

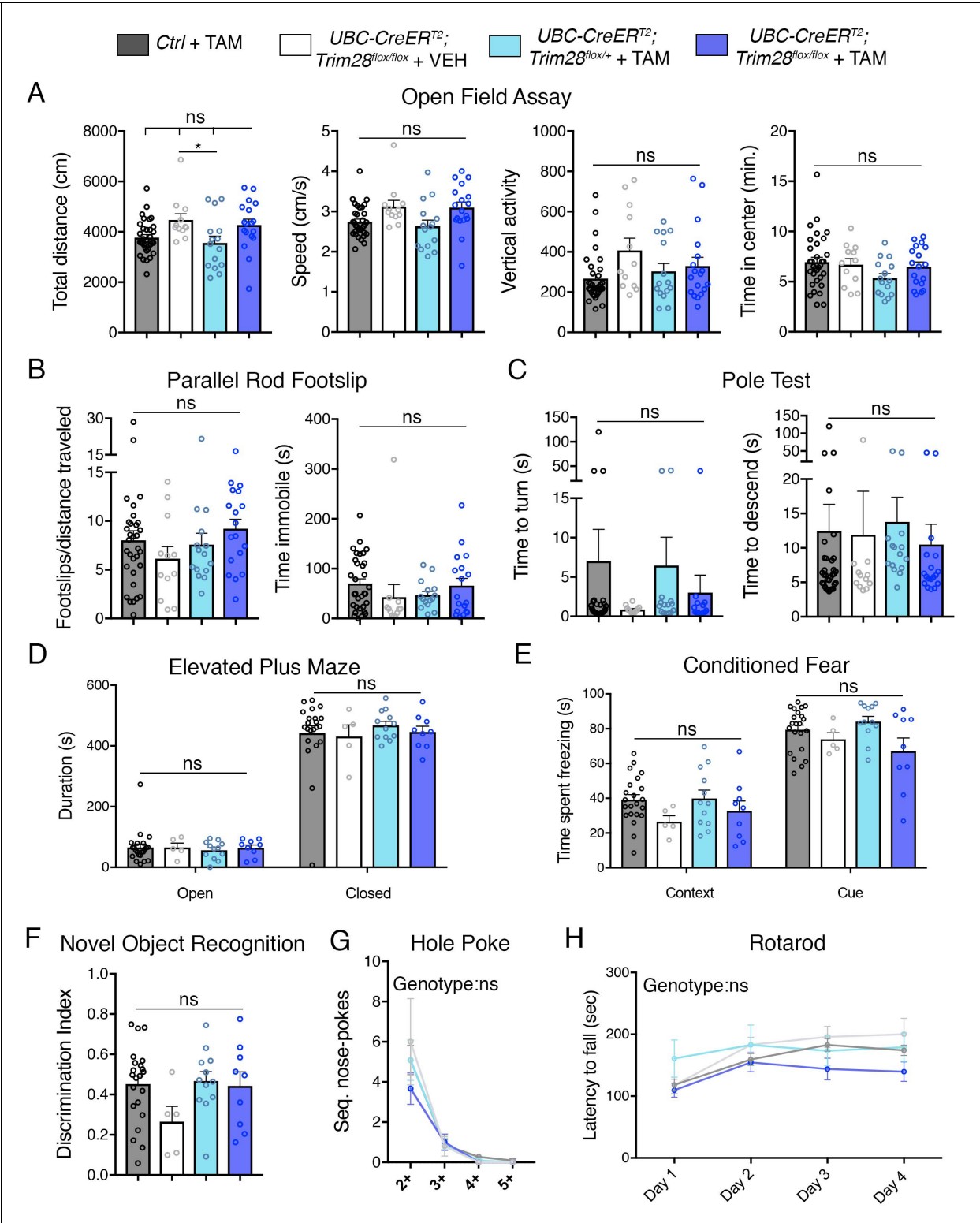

**Figure 3.** Adult depletion of Trim28 does not cause behavioral abnormalities. Adult knockout mice and littermate controls were subjected to: (A) Open field assay where total distance, speed, vertical activity and time in center were measured over a period of 30 min. (B) Parallel rod footslip analysis where number of footslips and time spent immobile were measured on a grid over a period of 10 min. (C) Pole test where the time to turn and descend were measured to a mouse on top (facing upward) of a 18' pole. (D) Elevated plus maze measured the time spent in open vs. closed arms during a period of 10 min. (E) Pavlovian conditioned fear analysis in both context and cued settings (day 2). (F) Novel object recognition assay showing the discrimination index for identifying the novel vs. familiar object. (G) Hole poke analysis of repetitive behavior measuring the number of sequential nose

*Figure 3 continued on next page*

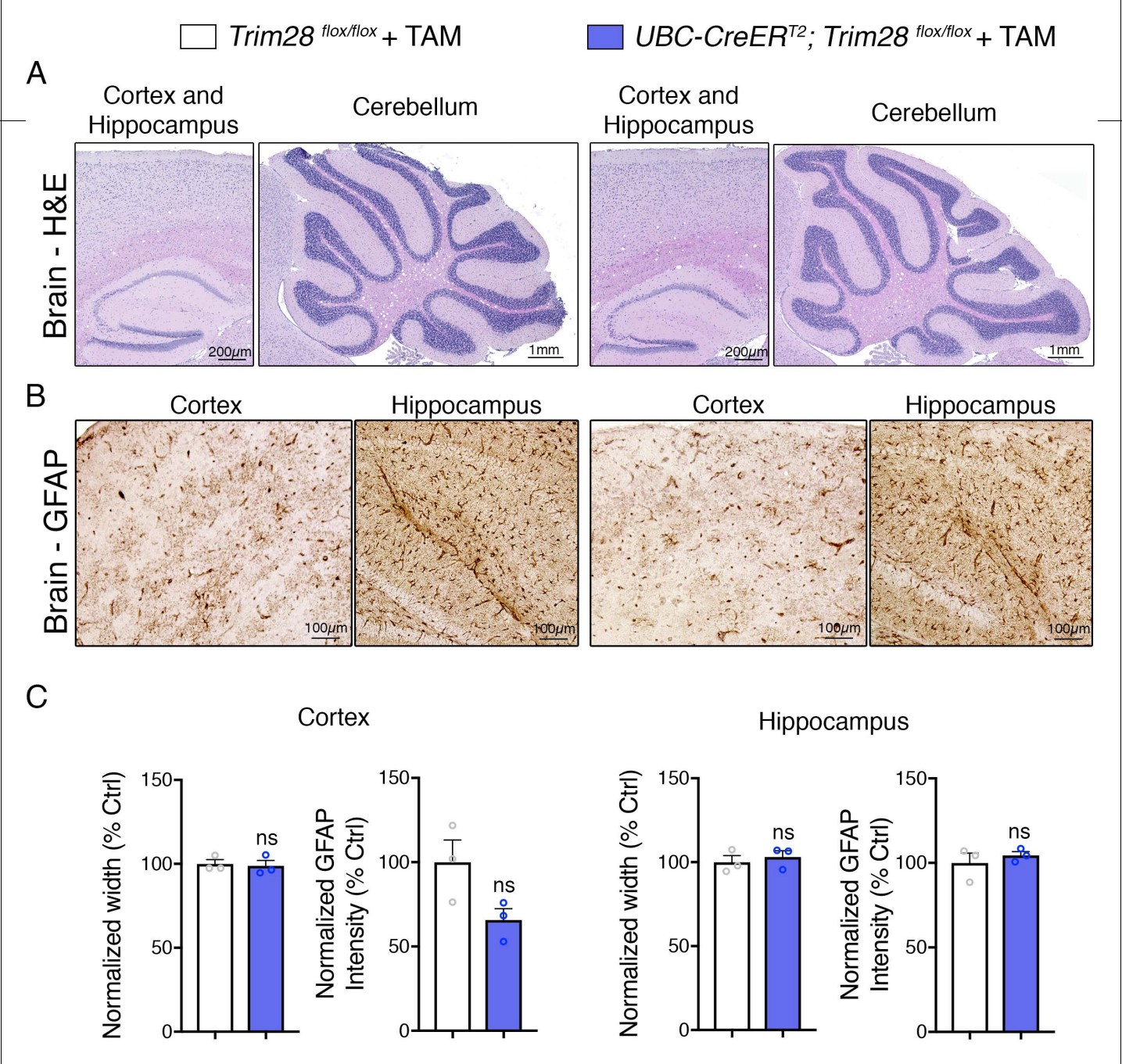

**Figure 4.** Adult depletion of Trim28 does not cause pathological abnormalities in the adult brain. Representative photomicrographs of the cortex, hippocampus and cerebellum stained with (**A**) H and E and (**B**) GFAP. (**C**) Quantification of cortical and hippocampal width as well as normalized GFAP intensity. For each test, $n$ = 3; ns denotes p>0.05.

DOI: https://doi.org/10.7554/eLife.36768.008

The following figure supplements are available for figure 4:

**Figure supplement 1.** Trim28 is expressed in the adult brain and can be effectively excised from adult mice.

DOI: https://doi.org/10.7554/eLife.36768.009

**Figure supplement 2.** Adult depletion of Trim28 does not cause peripheral pathological abnormalities.

DOI: https://doi.org/10.7554/eLife.36768.010

**Figure supplement 3.** Loss of Trim28 in adult mice does not disrupt global blood chemistry or iron homeostasis.

DOI: https://doi.org/10.7554/eLife.36768.011

inhibitor targeting TRIM28 may hold promise in the future with minimal side effects. Given the high expression of Trim28 in the adult mouse brain, additional characterization of these adult knockout animals will provide important insights into the potential of Trim28 downregulation in the context of disease; any physiological tradeoffs that may be incurred will thus be elucidated.

An important point of consideration moving forward into therapeutics is the mechanism by which TRIM28 regulates the steady state levels of α-Syn and tau. While our data suggest that TRIM28 forms a complex with α-Syn and tau (*Rousseaux et al., 2016*) and mediates their SUMOylation, we were not able to reconstitute this complex in a cell-free system, suggesting that other factors may be at play. Furthermore, while disruption of TRIM28 E3 ligase activity in vivo reduced α-Syn and tau levels, it likely did so by destabilization of TRIM28 itself. Thus, it is still unclear whether this inhibition represents a loss of enzymatic function or simply a structural loss. Further studies looking at the effect of this inhibition in adulthood or targeting other domains that may mediate TRIM28 SUMOyla-tion may hold promise. For instance, the bromodomain of TRIM28 could be an alternative target given that a mutation in cysteine 651 to an alanine (C651A) reduces its SUMOylation activity on another target, VPS34 (*Yang et al., 2013*). Alternatively, SUMOylation of α-Syn and tau via Trim28 may only be a partial or bystander effect. Additional studies in cell-free systems and in organisms will thus be crucial to look for factors that mediate this relationship to yield global mechanistic insight on regulation. Most importantly, this study highlights the importance of testing the loss of function of lethal variants in the adult. While databases such as ExAC and GnomAD (*Lek et al., 2016*) offer a window into the pathogenicity of variants in development, it should not be the only factor guiding target selection; especially for neurodegenerative conditions where treatment will often only occur in the later decades of life.

# Materials and methods

### Key resources table

| Reagent type (species) or resource | Designation | Source or reference | Identifiers | Additional information |
|---|---|---|---|---|
| Strain, strain background (*M. musculus*) | *Trim28^{E3MT}* (C66A, C69A, R72G) | This study | | Pure C57Bl/6J background |
| Strain, strain background (*M. musculus*) | *Trim28^{flox}* B6.129S2(SJL)-Trim28tm1.1Ipc/J | Jackson laboratory | Stock #018552 | Pure C57Bl/6J background |
| Strain, strain background (*M. musculus*) | UBC-CreERT2 B6.Cg-Ndor1 Tg(UBC-cre/ERT2)1Ejb/1J | Jackson laboratory | Stock #007001 | Pure C57Bl/6J background |
| Strain, strain background (*M. musculus*) | FVB/NCrl | Charles River | Code #207 | Pure C57Bl/6J background |
| Strain, strain background (*M. musculus*) | *Trim28^{+/-}* | *Rousseaux et al. (2016)*; this study | | Crossing Jax stock #018552 to #006054 |
| Strain, strain background (*M. musculus*) | *Snca^{-/-}* B6;129 × 1-Sncatm1 Rosl/J | Jackson laboratory | Stock #003692 | |
| Strain, strain background (*M. musculus*) | *Mapt^{-/-}* B6.129 × 1-Mapttm1 Hnd/J | Jackson laboratory | Stock #007251 | |
| Cell line (*H. sapiens*) | 293T | ATCC | CRL-3216 | |
| Cell line (*H. sapiens*) | 293T-shScram | This study; shScram from *Rousseaux et al. (2016)*. | | 293 T cells infected with retrovirus (pMSCV) harboring *shScramble*. Selected with 1 µg/mL of puromycin for at least 1 week before commencing experimentation |
| Cell line (*H. sapiens*) | 293T-shTRIM28 | This study; shTRIM28 from *Rousseaux et al. (2016)*. | | 293 T cells infected with retrovirus (pMSCV) harboring *shTRIM28*. Selected with 1 µg/mL of puromycin for at least 1 week before commencing experimentation |

*Continued on next page*

*Continued*

| Reagent type (species) or resource | Designation | Source or reference | Identifiers | Additional information |
|---|---|---|---|---|
| Transfected construct (*H. sapiens*) | *Flag-SUMO2* | This study | | |
| Transfected construct (*H. sapiens*) | *pKH3-HA-TRIM28* | Addgene | #45569 | |
| Transfected construct (*H. sapiens*) | *pKH3-HA-TRIM28-C65A/ C68A* | **Rousseaux et al. (2016)**; Addgene | #92199 | |
| Transfected construct (*H. sapiens*) | *pKH3* | Addgene | #12555 | |
| Transfected construct (*M. musculus*) | *AAV8-YFP-shScramble* | This study | accgcctgaagtctctgattaa | |
| Transfected construct (*M. musculus*) | *AAV8-YFP-shTrim28* | This study | ttgttgaactgtttgaacatgc | |
| Antibody | alpha-synuclein (C-20), Rabbit polyclonal | Santa Cruz Biotechnology | sc-7011-R | This antibody has been discontinued. |
| Antibody | alpha-synuclein (Clone 42), Mouse monoclonal | BD Biosciences | 610786 | |
| Antibody | Tau, Rabbit polyclonal | Dako | A0024 | |
| Antibody | Tau (Tau-5), Mouse monoclonal | Abcam | ab80579 | |
| Antibody | Trim28 (20C1), Mouse monoclonal | Abcam | ab22553 | |
| Antibody | SUMO2/3, Rabbit polyclonal | Abcam | ab3742 | |
| Antibody | Flag (M2), Mouse monoclonal | Sigma Aldrich | F1804 | |
| Antibody | UBC9, Goat polyclonal | Novus Biologicals | NB300-812 | |
| Antibody | Vinculin (hVIN-1), Mouse monoclonal | Sigma Aldrich | V9131 | |
| Antibody | GFAP (G-A-5), Mouse monoclonal | Sigma Aldrich | G3893 | |
| Sequence-based reagent (*M. musculus*), qPCR | *Mkrn3-f* | | ccatggagaaatatgcgaca | |
| Sequence-based reagent (*M. musculus*), qPCR | *Mkrn3-r* | | ctgagctgcatcccaagg | |
| Sequence-based reagent (*M. musculus*), qPCR | *Tcf5-f* | | tgatgcaatccggatcaa | |
| Sequence-based reagent (*M. musculus*), qPCR | *Tcf5-r* | | cacgtgtgttgcgtcagtc | |
| Sequence-based reagent (*M. musculus*), qPCR | *Pcdhb6-f* | | gccactagaagggctcgaat | |
| Sequence-based reagent (*M. musculus*), qPCR | *Pcdhb6-r* | | tgtctccacatctagctgcaa | |
| Sequence-based reagent (*M. musculus*), qPCR | *Klhdc4-f* | | cctggacaaaagttgacatcc | |
| Sequence-based reagent (*M. musculus*), qPCR | *Klhdc4-r* | | caaactccccaccgaagac | |
| Sequence-based reagent (*M. musculus*), qPCR | *Stac2-f* | | tgtctactagaaatcggtagccaag | |
| Sequence-based reagent (*M. musculus*), qPCR | *Stac2-r* | | agcgtcttgttctccacctg | |
| Sequence-based reagent (*M. musculus*), qPCR | *Smad3-f* | | ctcttggagcacatcctggt | |

*Continued on next page*

*Continued*

| Reagent type (species) or resource | Designation | Source or reference | Identifiers | Additional information |
| --- | --- | --- | --- | --- |
| Sequence-based reagent (*M. musculus*), qPCR | *Smad3-r* | | gcccagctggaaatatgc | |
| Sequence-based reagent (*M. musculus*), qPCR | *Cdkn1c-f* | | caggacgagaatcaagagca | |
| Sequence-based reagent (*M. musculus*), qPCR | *Cdkn1c-r* | | gcttggcgaagaagtcgt | |
| Sequence-based reagent (*M. musculus*), qPCR | *C1ql2-f* | | tcacgtaccacattctcatgc | |
| Sequence-based reagent (*M. musculus*), qPCR | *C1ql2-r* | | tgttgctggcgtagtcgta | |
| Sequence-based reagent (*M. musculus*), qPCR | *Snca-f* | | gaagacagtggagggagctg | |
| Sequence-based reagent (*M. musculus*), qPCR | *Snca-r* | | caggcatgtcttccaggatt | |
| Sequence-based reagent (*M. musculus*), qPCR | *Mapt-f* | | gagaatgccaaagccaagac | |
| Sequence-based reagent (*M. musculus*), qPCR | *Mapt-r* | | gtgagtccaccatgtcgatg | |
| Sequence-based reagent (*M. musculus*), qPCR | *Trim28-f* | | gctgctgccctgtctacatt | |
| Sequence-based reagent (*M. musculus*), qPCR | *Trim28-r* | | cacactggacaatccaccat | |
| Sequence-based reagent (*M. musculus*), qPCR | *S16-f* | | aggagcgatttgctggtgtgg | |
| Sequence-based reagent (*M. musculus*), qPCR | *S16-r* | | gctaccagggcctttgagatg | |
| Sequence-based reagent (*H. sapiens*), siRNA | *siScramble* | ThermoFisher Scientific | AM4611 | |
| Sequence-based reagent (*H. sapiens*), siRNA | *siUBC9* | ThermoFisher Scientific | AM16708-120322 | |
| Chemical compound, drug | Viomeillin | BioViotica | BVT-0359-C500 | |
| Chemical compound, drug | N-ethylmaleimide (NEM) | Sigma Aldrich | E3876-5G | |
| Chemical compound, drug | Tamoxifen | Sigma Aldrich | T5648-5G | |

## Cell culture

Cell culture was performed as previously described (*Rousseaux et al., 2016*). Briefly, HEK293T cells (ATCC CRL-3216; RRID:CVCL_0063, authenticated by manufacturer but not by researcher) devoid of mycoplasma were cultured in complete DMEM (DMEM +10% FBS+1 x antibiotic/antimycotic). Cells were plated in 6-well or 24-well plates for *SUMOylation assays* (see below) or siRNA and drug treatment, respectively. For the latter, cells were treated with 20 nM of indicated siRNAs or 10 μm Viomellein (or DMSO control) for 72 hr prior to lysis and western blot.

## SUMOylation assays

α-Syn and tau SUMOylation were assayed in cells as follows. Briefly, HEK293T cells were transfected with 3 μg Flag-SUMO2 and TRIM28 variants for 48 hr. Cells were harvested in cold PBS and spun down at 5,000 RPM for five minutes at 4°C. Cells were then lysed in SUMO lysis buffer (1% Triton X-100, 150 mM NaCl, 10 mM Tris pH 8.0, 10% glycerol, 20 mM N-ethyl maleimide and protease inhibitors [Roche]) for 40 min on ice with occasional vortexing. Cell debris were spun down at 15,000 RPM for 20 min at 4°C. Lysates were applied to Dynabeads (Protein G, 15 μL slurry) that were previously washed and then conjugated to 1 μg of antibody (α-Syn, C-20 Santa Cruz Biotechnology;

discontinued; tau, tau-5 Abcam; RRID:AB_304171) and incubated with rotation for 2 hr at 4°C. This sub-threshold pull-down allowed us to bypass the regulatory effect of TRIM28 on α-Syn and tau. Bound proteins were vigorously washed (to remove any interactors which themselves may be SUMOylated) four times in 500 μL of SUMO lysis buffer and eluted for 10 min at 95°C for downstream western blot analysis. For each condition, either cell lines stably knocking down *TRIM28* (*shTRIM28*) or non-silencing (*shScramble*) were used. In addition, TRIM28, TRIM28-Mut and control constructs were co-transfected at 300 ng per well (1:10 ratio to SUMO concentration). Alternatively, Flag-SUMO2 was pulled down using Flag-M2 magnetic beads (20 μl slurry, Sigma; RRID:AB_2637089) under denaturing conditions (first boiling the sample prior to the IP). Each SUMOylation assay was performed three independent times.

For the in vivo SUMOylation assay, mouse brains were harvested in RIPA buffer containing protease and phosphatase inhibitors (GenDepot). Samples were boiled for 5 min at 95°C, following which antibodies (2.5 μg) targeting α-Syn (C-20, SCBT) or Tau (Tau-5, Abcam) were incubated overnight with rotation at 4°C. Antibody-lysate complexes were bound to Dynabeads (25 μl, Protein G) for 2 hr at 4°C with rotation and then washed vigorously 5 × 1 mL in wash buffer (50 mM Tris pH 7.3, 170 mM NaCl, 1 mM EDTA, 0.5 % NP-40). Bound protein was eluted in Laemlli buffer at 85°C for 10 min. Lysates were run on SDS-PAGE followed by Western blot and SUMOylated species were detected by probing for SUMO2/3 (Abcam; RRID:AB_304041).

### Generation of Trim28^E3MT mice.

*Trim28^E3MT* mice on a pure C57Bl/6J background were generated via CRISPR/Cas9-mediated gene editing (*Wang et al., 2013*). Briefly, an sgRNA targeting the 5′ of *Trim28* was synthesized by direct PCR from pX330 (gift from Zhang lab, Addgene #42230) and in vitro transcribed with the MEGA-shortscript T7 Transcription kit (Invitrogen) using the following two primers (forward: 5′-TTAATAC-GACTCACTATAGGGCGTGTGTCGCGAGCGCCTGGTTTTAGAGCTAGAAATAGC-3′; reverse: 5′-AAAAGCACCGACTCGGTGCC-3′). A single stranded oligodeoxynucleotide (ssODN) was purchased from IDT for homologous-directed recombination introducing the C66A, C69A and R72G mutations in Trim28 (5′-CTGCAGCCGCGTCGTCCCCTGCGGGGGGCGGTGGCGAGGCGCAGGAGCTTTTA-GAACATGCCGGTGTCGCCAGGGAAGGACTCAGACCAGAACGGGATCCTCGGCTGCTGCCCTG TCTACATTCGGCCTGCAGTGCCTGCCTGGGCCCCGCTACACCCGCCGCAGCGAATAA TTCGGGGGATGGCGGCTCGG-3′). The PAM (protospacer adjacent motif) and additional adjacent synonymous mutations were introduced to increase editing efficiency and allow for simple genotyping by differential primer hybridization. On the day of injection, Cas9 protein (PNA Bio), sgRNA and repair template (ssODN) were injected (pronuclear) into ova from C57Bl/6 female mice and transferred into oviducts of pseudopregnant females. The following primers were used to distinguish the E3 mutant allele (forward: 5′-TTGGCGGCGAGCGCACTTGC-3′; reverse: 5′-CCCTGGCGA-CACCGGCATG-3′ or forward: 5′-CATGCCGGTGTCGCCAGGGA-3′; reverse: 5′-TCCCACAGGACA TACCTGGTTAGCATCCTGG-3′) from the wildtype allele (forward: 5′-TTGGCGGCGAGCGCAC TTGC-3′; reverse: 5′-TCGCGACACACGCCGCAGTG-3′ or 5′-CACTGCGGCGTGTGTCGCGA-3′; reverse: 5′-TCCCACAGGACATACCTGGTTAGCATCCTGG-3′). Founder mice were backcrossed at least three times prior to experimentation to get rid of potential off-target mutations.

### Tamoxifen injections

Tamoxifen injections were performed as previously described (*Sztainberg et al., 2015*). Briefly, starting at 8–12 weeks of age, tamoxifen or vehicle (peanut oil) was injected intraperitoneally at a dose of 100 mg/kg, three times a week for four weeks. Mice were left to recover for at least two weeks before proceeding with behavioral, biochemical and histological assessment.

### AAV generation and P0 injections

An AAV8 vector containing both YFP and a miRE cassette-containing shRNA (*Fellmann et al., 2013*) under the control of the chicken beta actin (CBA) promoter was generated using Gibson cloning. Individual shRNA sequences were generating using the splaSH algorithm (*Pelossof et al., 2017*). Each shRNA vector was tested for efficiency in Neuro2A cells prior to virus generation.

AAV delivery was carried out in neonatal (P0) FVB mouse pups as previously described (*Kim et al., 2013*). Briefly, neonatal pups (<8 hr from birth) were separated from lactating dams and

anesthetized on ice. $1 \times 10^{11}$ viral genomes were injected per ventricle (total of $2 \times 10^{11}$ genomes per mouse) and mice were left to recover on a heated pad before returning them to their mother. Tissue from the caudal region of the cerebrum (cortex + hippocampus) was harvested ten weeks post injection as this region had the maximal viral expression (YFP positive signal) and offered optimal *Trim28* knockdown by qPCR. RNA extraction was performed using the RNeasy mini kit (Qiagen).

## Behavioral analysis

Behavioral analysis was performed by an experimenter blind to the treatment and genotype of the animals. Animal behavior was conducted between 10 am and four pm for each test and was carried out when the animals were 14–22 weeks old (6–10 weeks post tamoxifen injection). The open-field analysis (*Lu et al., 2017*), parallel rod footslip (*Ure et al., 2016*), pole test (*Rousseaux et al., 2012*), elevated plus maze (*Lu et al., 2017*), conditioned fear (*Lu et al., 2017*), novel object recognition (*Antunes and Biala, 2012*), hole poke (*Ito-Ishida et al., 2015*) and rotarod (*Lasagna-Reeves et al., 2015*) were performed as previously described. For each test, mice were left to habituate in the testing room with ambient white noise for 30–60 min prior to testing.

## Histological analysis

For frozen sections: Free floating sections (25 µm) were mounted and dried on polarized slides (>48 hr). Slides were then stained for Cresyl violet and GFAP (RRID:AB_2314539) as previously described (*Rousseaux et al., 2016*). For GFAP quantification, photomicrographs were taken using the 10x objective on a Leica DM4000 LED. The percentage of immunoreactive area for GFAP was calculated using ImageJ. Briefly, each DAB-stained image was converted to 8-bit greyscale and made into a binary image using a threshold cutoff of 10% for a representative WT section (after which, the same settings were used for all of the sections in question). Area of interest (Hippocampus or Cortex) was outlined and total area was measured. Within this area, the 'Analyze particles' function was used to determine the area of each outlined immunoreactive entity. The sum of these entities was set at the GFAP positive area and the percentage immunoreactive area was presented as GFAP positive area compared to total area (in %). For cresyl violet staining, the relative width of either the caudal cortex or the CA1 region of the hippocampus was measured in four independent sections.

For paraffin-embedded sections: Formalin-fixed tissues were embedded in paraffin and sectioned on a microtome at 5 µm thickness. Sections were deparaffinized in a series of xylene and ethanol washes before being subjected to antigen retrieval for 10 min at 95°C in a buffer containing 10 mM sodium citrate and 0.02% Tween (pH 6.0). Sections were then blocked for one hour at room temperature in PBS + 0.3% Triton X-100 and 5% FBS and stained in blocking buffer containing either 1:400 anti-GFAP (GA5, Sigma) or 1:500 anti-Trim28 (20C1, Abcam) and corresponding secondary antibodies (Vectastain mouse elite ABC kit or Donkey anti-mouse Alexa 488 secondary; RRID:AB_2341099). Fluorescent sections were counterstained using DAPI. Gross morphology was assessed by performing hematoxylin and eosin (H and E) staining using standard protocols.

## Mouse blood collection

Mice were anaesthetized with isoflurane and blood was collected from the retro-orbital sinus. A capillary was inserted into the medial canthus of the eye of the anaesthetized mouse. Applying a slight pressure to the capillary allows the blood flow to be directed to a collection tube. After letting the blood coagulate for 30 min, the serum is collected post centrifugation 4 min at 14,000 r.p.m. for analyte analysis with Charles River Laboratories. qPCR analysis qPCR was performed as previously described (*Rousseaux et al., 2016*). Briefly, 1 µg of RNA isolated from mouse tissue (cortex, hippocampus, midbrain, heart, liver and spleen) was reverse transcribed into cDNA. qPCR primers were designed to span exons to prevent gDNA contamination and can be found in the Key Resources Table. We used the ddCT method as well as S16 as a loading control to calculate relative transcript abundance. Due to the multiple transcript measurements, we used multiple t-tests with an FDR correction of 10% to avoid false discoveries.

## Statistical analysis

Experimental analysis was performed in a blinded manner when possible. Statistical tests were performed in accordance with the experimental design. For instance, for simple comparisons we used

Student's t-test whereas multi-group analyses we used one- or two-way ANOVA followed by a post-hoc test. For comprehensive statistical coverage of each experiment throughout this manuscript, please see *Supplemental file 1*. In each case, *, **, ***, **** and ns denote $p < 0.05$, $p < 0.01$, $p < 0.001$, $p < 0.0001$ and $p > 0.05$, respectively.

## Acknowledgements

The authors thank members of the Zoghbi lab for important discussions and critical feedback on the manuscript, LA. Lavery for helping with TRIM28 structural modeling experiments and A Hatcher and J Noebels for the *Mapt*[-/-] mice. This research was supported in part by the Robert A and Renée E Belfer Family Foundation, the Huffington Foundation, The Hamill Foundation, the Howard Hughes Medical Institute and UCB Pharma (HYZ.), the Parkinson's Foundation Stanley Fahn Junior Faculty Award PF-JFA-1762 (MWCR.), the behavior, pathology, RNA in situ hybridization and confocal cores at the Jan and Dan Duncan Neurological Research Institute and the BCM Intellectual and Developmental Disabilities Research Center (NIH U54 HD083092 from the Eunice Kennedy Shriver National Institute of Child Health and Human Development). The IDDRC Microscopy Core was used for this project. The content is solely the responsibility of the authors and does not necessarily represent the official views of the Eunice Kennedy Shriver National Institute of Child Health and Human Development or the National Institutes of Health.

## Additional information

### Competing interests

Huda Y Zoghbi: Senior editor, *eLife*. The other authors declare that no competing interests exist.

### Funding

| Funder | Grant reference number | Author |
|---|---|---|
| Parkinson's Foundation | Stanley Fahn Junior Faculty Award (PF-JFA-1762) | Maxime WC Rousseaux |
| UCB Pharma | | Huda Y Zoghbi |
| Robert A. and Renee E. Belfer Family Foundation | | Huda Y Zoghbi |
| The Huffington Foundation | | Huda Y Zoghbi |
| The Hamill Foundation | | Huda Y Zoghbi |
| Howard Hughes Medical Institute | | Huda Y Zoghbi |
| Intellectual and Developmental Disabilities Research Center | NIH U54 HD083092 | Huda Y Zoghbi |

The funders had no role in study design, data collection and interpretation, or the decision to submit the work for publication.

### Author contributions

Maxime WC Rousseaux, Conceptualization, Resources, Data curation, Formal analysis, Supervision, Funding acquisition, Validation, Investigation, Visualization, Methodology, Writing—original draft, Project administration; Jean-Pierre Revelli, Data curation, Formal analysis, Validation, Methodology, Writing—review and editing; Gabriel E Vázquez-Velez, Formal analysis, Validation, Investigation, Methodology, Writing—review and editing; Ji-Yoen Kim, Data curation, Validation, Investigation, Methodology; Evelyn Craigen, Kristyn Gonzales, Jaclyn Beckinghausen, Data curation, Investigation, Methodology; Huda Y Zoghbi, Conceptualization, Resources, Supervision, Funding acquisition, Writing—original draft, Project administration

## Author ORCIDs

Maxime WC Rousseaux  http://orcid.org/0000-0002-2737-6193
Huda Y Zoghbi  http://orcid.org/0000-0002-0700-3349

## Ethics

Animal experimentation: Up to five mice were housed per cage and kept on a 12 h light; 12 h dark cycle and were given water and standard rodent chow ad libitum. All procedures carried out in mice were approved by the Institutional Animal Care and Use Committee for Baylor College of Medicine and Affiliates under protocol AN-1013.

## Decision letter and Author response

Decision letter https://doi.org/10.7554/eLife.36768.016
Author response https://doi.org/10.7554/eLife.36768.017

## Additional files

### Supplementary files

• Supplemental file 1. Detailed statistical analysis for all data throughout the manuscript.
DOI: https://doi.org/10.7554/eLife.36768.012

• Transparent reporting form
DOI: https://doi.org/10.7554/eLife.36768.013

### Data availability

No datasets were generated in this study. All data are presented in this manuscript.

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
