## [Decision Letter]

Thank you for submitting your article "Depleting Trim28 in adult mice is well tolerated and reduces levels of α-synuclein and tau" for consideration by *eLife*. Your article has been reviewed by 3 peer reviewers, including a Reviewing Editor, and the evaluation has been overseen by a Senior Editor. The following individuals involved in review of your submission have agreed to reveal their identity: Patrik Verstreken (Reviewer #2); Ming Guo (Reviewer #3).

The reviewers have discussed the reviews with one another and the Reviewing Editor has drafted this decision to help you prepare a revised submission.

The manuscript by Rousseaux et al. is a follow up study to the same group's *eLife* paper in 2016. In the previous work, Rousseaux demonstrated that the E3 ligase TRIM28 regulates levels of α-synuclein and tau in HEK293T cells, primary mouse neurons and in the mouse hippocampus (via injection of shRNA). Loss of TRIM28 also decreased tau levels in an overexpression fly model and mitigated neurodegenerative phenotypes. Similar findings were observed upon reduction of TRIM28 (via shRNA) in an α-synuclein mouse model. Overexpression of TRIM28 lead to toxic nuclear accumulations of α-synuclein and tau while overexpression of a catalytically inactive TRIM28 (C65A/C68A), while less stable, did not.

In this manuscript, Rousseaux et al. demonstrate that inhibiting UBC9, the sole SUMO ligase, decreased α-synuclein and tau levels. Furthermore, overexpression of TRIM28 increased the SUMOylation of these proteins but expression of C65A/C68A TRIM28 does not. The authors characterized knockin mice heterozygous for the C65A/C68A mutation and found that Trim28 was decreased as were α-synuclein and tau levels. Mice homozygous for this mutation (or for a knockout mutation in Trim28) resulted in embryonic lethality. The authors used both AAV-delivered shRNA and conditional deletion via UBC-CreER to decrease Trim 28 in the postnatal brain. No overt morphological or behavioral phenotypes were observed in these mice suggesting that decreasing Trim28 levels after development might be a safe therapeutic means of decreasing α-synuclein and tau and warrant further investigation.

While the reviewers believe the manuscript is interesting and a valuable continuation of their previous paper, some revisions are necessary prior to publication.

1) The hypothesis that TRIM28 regulates α-synuclein and tau levels through sumoylation. It is not stated clearly exactly how sumoylation would bring about this regulation. The story with TRIM28 sumo ligase activity appears to be more complicated than indicated in the current manuscript. Trim28 sumo ligase activity has been localized to both the N-terminal Ring domain (mutated in the current and previous paper), and to the C-terminal PHD domain. The latter domain has been implicated in auto-sumoylation as well as sumoylation of other substrates. RING domains can also mediate protein-protein interactions. Thus, experiments in which the RING domain is mutated do not necessarily show that TRIM28 is directly bringing about sumoylation via this domain. It may be a protein-protein interaction domain, or act in other indirect ways.

It is not clear from these experiments (Figure 1B) that the total levels of α-synuclein or tau are significantly altered. In short, while regulation of TRIM28 levels does result in changes in the levels of sumoylated α-synuclein and tau, it is less clear (based on Figure 1bB and C) that total levels are altered. Often sumoylation effects in cells seem to be regulatory, in which a modest fraction of the target protein is modified and then plays a regulatory role, rather than the bulk of the protein being modified, as with ubiquitylation that leads to proteasome-dependent degradation. Thus, I would suggest that the authors consider in their Discussion a variety of models in which the effects of altering TRIM28 activities (binding and ligase activities) are considered together with other models in which sumoylation occurs, but may not necessarily be the primary mechanism by which α-synuclein and tau are being tagged for elimination.

2) The authors state "trim28 has several important functions throughout the cell…" It would be good to include some references, since it has in fact been implicated in a number of important processes. These become important later in the context of considering the consequences of adult-specific downregulation or knockout.

3) The authors state "our study suggests that depletion of more than 75% of total TRIM28 from the body is safe". "Safe" seems to be too strong. The authors might consider something more nuanced such as "our study failed to identify deleterious phenotypes on depletion of more than 75% of total TRIM28 from the body, as determined using a number of assays in multiple tissues. These findings suggest that inhibition of TRIM28 in adulthood may provide a strategy for.… with minimal side effects. TRIM28 has been described to function in a number of contexts, ranging from transposon mobilization, transcriptional regulation, etc… Further characterization of these adult KO animals will provide important insights into the potential of trim28 downregulation to inhibit disease, and any physiological tradeoffs that may be incurred." Similarly in the last paragraph of the Results and Discussion. Reference to muscle specific loss of trim28 as safe. Again, "safe" may not be the most appropriate word. The paper referenced only notes there is no effect on adiposity. No other data are presented.

4) A description of the statistical tests used to analyze the data need to be added.

5) Scale bars should be added to histological images.

---

## [Author Response]

[…] While the reviewers believe the manuscript is interesting and a valuable continuation of their previous paper, some revisions are necessary prior to publication.1) The hypothesis that TRIM28 regulates α-synuclein and tau levels through sumoylation. It is not stated clearly exactly how sumoylation would bring about this regulation. The story with TRIM28 sumo ligase activity appears to be more complicated than indicated in the current manuscript. Trim28 sumo ligase activity has been localized to both the N-terminal Ring domain (mutated in the current and previous paper), and to the C-terminal PHD domain. The latter domain has been implicated in auto-sumoylation as well as sumoylation of other substrates. RING domains can also mediate protein-protein interactions. Thus, experiments in which the RING domain is mutated do not necessarily show that TRIM28 is directly bringing about sumoylation via this domain. It may be a protein-protein interaction domain, or act in other indirect ways.

We have further expanded on our rationale for studying α-Syn and tau SUMOylation over other potential effects in the first paragraph of the Results and Discussion. In hindsight, we agree that the relationship between TRIM28 and α-Syn and tau is likely not as clear-cut as it seems (Results and Discussion, second paragraph). We have therefore highlighted this at the end of our Discussion (subsection “SUMOylation assays”, first paragraph).

It is not clear from these experiments (Figure 1B) that the total levels of α-synuclein or tau are significantly altered. In short, while regulation of TRIM28 levels does result in changes in the levels of sumoylated αB and C) that total levels are altered. Often sumoylation effects in cells seem to be regulatory, in which a modest fraction of the target protein is modified and then plays a regulatory role, rather than the bulk of the protein being modified, as with ubiquitylation that leads to proteasome-dependent degradation. Thus, I would suggest that the authors consider in their Discussion a variety of models in which the effects of altering TRIM28 activities (binding and ligase activities) are considered together with other models in which sumoylation occurs, but may not necessarily be the primary mechanism by which α-synuclein and tau are being tagged for elimination.

To this second point, we wish to clarify that the amounts of α-Syn and tau that are pulled down in these assays (Figure 1B and C) are non-saturating. Therefore, when looking at the IP conditions, one does not expect to see changes in α-Syn and tau levels following TRIM28 modulation. These “sub-threshold pulldowns” allow for easier interpretation of SUMOylation data (the numerator) as the candidate immunoprecipitated (the denominator) is constant. Thus, from these results, we cannot comment as to the effect on α-Syn and tau levels but instead we can only comment on the effect of TRIM28 on their SUMOylation status.

2) The authors state "trim28 has several important functions throughout the cell…" It would be good to include some references, since it has in fact been implicated in a number of important processes. These become important later in the context of considering the consequences of adult-specific downregulation or knockout.

We have included references that highlight the multiple functions of TRIM28 in the cell.

3) The authors state "our study suggests that depletion of more than 75% of total TRIM28 from the body is safe". "Safe" seems to be too strong. The authors might consider something more nuanced such as "our study failed to identify deleterious phenotypes on depletion of more than 75% of total TRIM28 from the body, as determined using a number of assays in multiple tissues. These findings suggest that inhibition of TRIM28 in adulthood may provide a strategy for.… with minimal side effects. TRIM28 has been described to function in a number of contexts, ranging from transposon mobilization, transcriptional regulation, etc. Further characterization of these adult KO animals will provide important insights into the potential of trim28 downregulation to inhibit disease, and any physiological tradeoffs that may be incurred." Similarly in the last paragraph of the Results and Discussion. Reference to muscle specific loss of trim28 as safe. Again, "safe" may not be the most appropriate word. The paper referenced only notes there is no effect on adiposity. No other data are presented.

We agree that without complete longitudinal characterization of these mice, using the term “safe” may be premature. Thus, we have reworded the text to reflect our observations as per the reviewers’ suggestions (Results and Discussion, seventh paragraph).

4) A description of the statistical tests used to analyze the data need to be added.

We have now included a brief description of the statistical tests used in the Materials and methods and have now incorporated all statistical tests used throughout the study as Supplementary file 1.

5) Scale bars should be added to histological images.

These have been added.